# The Beneficial Effects of Triterpenic Acid and Acteoside in an In Vitro Model of Nonalcoholic Steatohepatitis (NASH)

**DOI:** 10.3390/ijms23073562

**Published:** 2022-03-24

**Authors:** Noel Salvoza, Chiara Bedin, Andrea Saccani, Claudio Tiribelli, Natalia Rosso

**Affiliations:** 1Fondazione Italiana Fegato—ONLUS, Area Science Park, Basovizza SS14 km 163.5, 34149 Trieste, Italy; noel.salvoza@fegato.it; 2Philippine Council for Health Research and Development, DOST-Bicutan, Taguig City 1631, Philippines; 3ABResearch S.R.L., Via dell’Impresa 1, 36040 Brendola, Italy; chiara.bedin@abres.it (C.B.); andrea.saccani@abres.it (A.S.)

**Keywords:** NAFLD, NASH, triterpenic acid, acteoside, in vitro models

## Abstract

Triterpenic acid (TA) and acteoside (ACT), the major components of APPLIVER and ACTEOS, respectively, have been reported to exert hepatoprotective effects, but the molecular mechanisms remain elusive, particularly in the NAFLD/NASH context. We assessed their effects in our well-established in vitro model resembling the pathophysiological mechanisms involved in NASH. Human hepatocytes and hepatic stellate cells were exposed to free fatty acids (FFA) alone or in combination with APPLIVER and ACTEOS as a mono- or co-culture. Steatosis, inflammation, generation of reactive oxygen species (ROS), and collagen deposition were determined. ACTEOS reduced both the TNF-α and ROS production, and, most importantly, attenuated collagen deposition elicited by the excess of FFA in the co-culture model. APPLIVER also showed inhibition of both TNF-α production and collagen deposition caused by FFA accumulation. The compounds alone did not induce any cellular effects. The present study showed the efficacy of APPLIVER and ACTEOS on pathophysiological mechanisms related to NASH. These in vitro data suggest that these compounds deserve further investigation for possible use in NASH treatment.

## 1. Introduction

NAFLD is the most common chronic liver disease worldwide with an estimated prevalence of 24–25% in the general population [1]. In the coming years, NAFLD is projected to be the main indication of liver transplantation, surpassing hepatitis C infection [2,3]. NAFLD is tightly associated with obesity and other features of metabolic syndrome (MS). For this reason, NAFLD is now considered as the hepatic manifestation of MS [4,5].

NAFLD is a wide-spectrum disease that comprises two phenotypes: non-alcoholic fatty liver (NAFL) is defined by simple steatosis (SS); and nonalcoholic steatohepatitis (NASH) is characterized by cellular injury induced by SS. NAFLD is accompanied by many pathophysiological mechanisms, such as sustained inflammation, oxidative stress, extracellular matrix remodeling, and hepatic stellate cell (HSC) activation [6]. These mechanisms added to the complexity of the proposed “multiple-hit model” of NAFLD pathogenesis, involving the interaction of genetic and environmental factors as well as the crosstalk between different organs. However, the onset of the disease is still represented by the accumulation of fat in the liver [5,6].

The turning point for NASH progression is the activation of HSCs, which can lead to fibrosis, considered the most important morbidity and prognostic factor in NAFLD development [7]. Fibrosis triggers dysregulation in extracellular matrix production, which in turn initiates the release of noxious stimuli involved in inflammation and the progression of NAFL to NASH [8,9]. The development of inflammation and fibrosis in NASH underlies cirrhosis and hepatocellular carcinoma (HCC) [10].

To date, despite many efforts to unravel NAFLD pathogenesis, no pharmacological treatment for this disease is currently approved. Drugs available are often used to target the co-morbidities or the specific disease mechanisms involved in NAFLD. Therefore, preclinical studies are important for the understanding of NAFLD pathophysiology and future drug development.

Triterpenic acid (TA) is a triterpene isolated from various plants belonging to Rosaceae family, mostly from the leaves and whole herbs [11,12]. This compound was found to possess various pharmacological properties, including hepatoprotective effects [13,14]. On the other hand, the hepatoprotective effects of acteoside (ACT), a phenylpropanoid present in various species of the order Lamiales, were evaluated in LPS- and carbon tetrachloride (CCl_4_)-induced acute hepatic damage in mice. Triterpenic acid and acteoside ameliorated the elevation of serum aspartate transaminase (AST) and alanine aminotransferase (ALT), as well as the depletion of endogenous antioxidants and oxidative stress markers [15,16]. Despite substantial evidence regarding the general hepatoprotective effects of both compounds in preclinical settings, the molecular mechanisms underlying the beneficial effects are still elusive.

Several models have been developed in an attempt to mimic the molecular mechanisms involved in the onset of NAFLD and the progression of NASH [17]. Among these, two-dimensional cell culture models (monoculture and co-culture) provide the most advantages, such as controlled conditions, are easy to handle, and offer reliable information on NASH’s development [17,18,19]. These characteristics make the system suitable for preclinical testing to evaluate drug efficacy and cellular tolerance.

We previously developed an in vitro human NASH model able to reproduce the initial phases of NASH’s development due to cell-to-cell interactions. Specifically, we reported that excess FFA accumulation in hepatocytes leads to increase inflammatory response, oxidative stress, and activation of HSCs, which are all key hallmarks of NASH [9,19,20].

In the present study, we assessed the effects of APPLIVER (*Cydonia oblonga* cell extract, expressed in triterpenic acid) and ACTEOS (*Lippia citriodora* cell extract, expressed in acteoside) in our in vitro model resembling the pathophysiological mechanisms involved in NASH. We demonstrated that APPLIVER and ACTEOS are promising therapeutic compounds for counteracting inflammation and ROS generation caused by an excess of FFA in the hepatocytes. Most importantly, both compounds attenuated collagen production elicited by the excess of FFA in the co-culture model.

## 2. Results

### 2.1. Determination of Experimental Concentration

To assess the safety of the compounds in a Huh7 monoculture and simultaneous co-culture (SCC) with or without FFA, a cell viability test (MTT assay) was performed for both compounds. MTT assay was also used to determine the effective treatment concentration of each compound needed for the succeeding experiments. Briefly, the monoculture and SCC were exposed to 1200 μM of FFA in the presence of increasing concentrations of the compounds at specific time points (24 h for Huh7; 96 and 144 h for SCC). Cells were also exposed to a medium alone with increasing concentrations of compounds to assess their toxicity in the monoculture and SCC without FFA.

The extent of cytotoxicity from each concentration of the compound was quantified as a percentage of cell viability relative to the vehicle control. Pursuant to ISO 10993-5 (in vitro cytotoxicity test on extracts), percentages of cell viability above 80% are considered as non-cytotoxic; within 60–80% they are considered weakly, within 40–60% they are considered moderately, and below 40% they are considered strongly cytotoxic [20]. Results are shown in Table 1. No cytotoxicity was observed in the Huh7 monoculture and SCC at 50 nM APPLIVER and at 100 nM ACTEOS in all experimental conditions. We then selected 50 nM APPLIVER and 100 nM ACTEOS as the test concentrations.

### 2.2. Effects of APPLIVER and ACTEOS on Fat Accumulation

The content of intracellular lipid droplets was determined by Nile red staining through flow cytometry. As shown in Figure 1a FFA induced a 203 ± 35% (*p* < 0.001) increase in fat accumulation vs. vehicle-treated cells in the Huh7 monoculture, similar to our previous data [21]. The co-treatment with both compounds did not alter the intracellular fat content. Likewise, the addition of both compounds in the absence of FFA did not induce steatosis (Figure 1b).

We then assessed the effect of the compounds on the intracellular fat content of the SCC system at a longer exposure time (96 h). A 96 h FFA exposure induced a 214 ± 11% (*p* < 0.01) increase in intracellular fat vs. the vehicle-control; as for the 24 h exposure, the intracellular fat accumulation was not changed by either of the two compounds (Figure 2a). The addition of both compounds in the absence of FFA did not induce steatosis on SCC (Figure 2b).

### 2.3. Effects of APPLIVER and ACTEOS on the Inflammatory Response

To assess the anti-inflammatory effect of APPLIVER and ACTEOS, we determined the gene expression of the pro-inflammatory cytokines IL-6, IL-8, and TNF-α. FFA induced an upregulation of the inflammatory mediators, but only IL-8 and TNF-α were statistically significant (Figure 3). Both APPLIVER and ACTEOS induced a significant (*p* < 0.05) downregulation of the TNF-α gene expression. Moreover, neither of the compounds in the absence of FFA changed the expression of the cytokines (Appendix A).

The effect of the compounds to a TNF-α release in cell culture supernatant was determined by ELISA (Figure 4) after 1 h of treatment, since it has been reported that the peak concentrations of TNF-α in the cell supernatant usually occurs at 1–2 h [22,23,24]. In line with the gene expression results, APPLIVER and ACTEOS significantly reduced the release of TNF-α. In the absence of FFA, neither treatment with APPLIVER nor with ACTEOS induced a TNF-α release in the cell culture supernatant (Appendix A).

### 2.4. Effects of APPLIVER and ACTEOS on the Production Reactive Oxygen Species

The antioxidant properties of APPLIVER and ACTEOS alone or in combination with FFA were evaluated on a Huh7 monoculture after 1 h of treatment. The exposure to FFA induced a 40% (*p* < 0.001) increase in ROS generation vs. the vehicle-treated control, which was similar to hydrogen peroxide, the positive control (Figure 5). In the presence of FFA, APPLIVER reduced ROS generation, though the difference was not statistically significant, while ACTEOS significantly (*p* < 0.05) reduced ROS production. The exposure to APPLIVER and ACTEOS in the absence of FFA did not induce any change in the cellular redox state (Appendix A).

### 2.5. Effects of APPLIVER and ACTEOS on Collagen (COL1A1) Production of SCC

The production of fibrillar type-I collagen is associated with ECM remodeling and liver fibrosis. We evaluated the effects of APPLIVER and ACTEOS on the expression of collagen type I alpha 1 chain (COL1A1). The gene expression of COL1A1 was increased in SCC after 96 h and even more after 144H of FFA exposure (Figure 6). APPLIVER showed a not significant reduction in COL1A1 expression both at 96 h and 144 h, while ACTEOS significantly (*p* < 0.05) reduced the expression by 50% vs. FFA at 96 h; this effect was not observed at 144 h.

At the protein level (Figure 7), treatment with either APPLIVER or ACTEOS showed a 60% reduction in collagen at 96 h vs. FFA (*p* < 0.05). This effect was present also at the longest time tested (144 h). Exposure to APPLIVER and ACTEOS only, in the absence of FFA, did not affect the gene expression or extracellular collagen deposition at both time points (Appendix A).

## 3. Discussion

APPLIVER and ACTEOS are two biotechnological compounds obtained by an innovative biotechnological platform of plant cell culture [25]. Triterpenic acid, the main component of APPLIVER, is isolated from various plants such as *Vochysia divergens*, *Potentilla chinensis*, *Cydonia oblonga*, and *Malus domestica* [26,27]. Mounting evidence suggests that TA exerts several biological activities, including anticancer, anti-inflammatory, antidiabetic, and antihepatotoxic properties [13,28,29,30]. Acteoside, also known as verbascoside, is the major component of ACTEOS. Acteoside is a phenylethanoid glycoside ingredient that can be found in more than 200 plant species, including *Plantago* and *Lippia* species [31]. As for TA, acteoside has been reported to exert various pharmacological effects, such as antioxidant, antimicrobial, anti-inflammatory, neuroprotective, anticancer, and hepatoprotective effects [32,33].

Both compounds showed hepatoprotective effects on chemically-induced liver damage in mice by lowering transaminase levels and improving total bilirubin levels [15]. Additionally, both compounds were found to reduce histological alterations in liver tissues [15,27]. However, little evidence is found in the literature when it comes to their effects in NAFLD/NASH-related pathophysiological mechanisms such as steatosis, oxidative stress, inflammation, and fibrosis. The marginal data regarding the molecular mechanisms hamper drug development, particularly in the context of NAFLD/NASH.

In this study, we assessed the effects of the compounds in counteracting the cellular events following lipid accumulation in both hepatocytes and a simultaneous co-culture (hepatocytes-HSCs). Several models of NASH propose that the FFA accumulation in hepatocytes results in hepatocyte injury, inflammation, and oxidative stress [17,18]. In particular, persistent inflammation in hepatocytes may trigger the activation of HSCs, which in turn are responsible for the production and deposition of an extracellular matrix such as collagen, which is a hallmark of NASH [34,35].

We developed an in vitro model of NASH where the exposure of hepatocytes to high concentrations of FFA promotes inflammation, oxidative stress, and fibrogenic response similar to those observed in patients with NASH [21]. Moreover, our group also developed an in vitro model that reproduces the initial events involved in fibrosis with the overexpression of alpha-smooth muscle actin (α-SMA), the accumulation of extracellular collagen, and the modulation of metalloproteinases (MMPs) and the tissue inhibitor of metalloproteinases (TIMP) [36].

An FFA overload in hepatocytes results in inflammation. Although resident liver macrophages (Kupffer cells) are important drivers of hepatic inflammation, hepatocytes per se have a role in initiating the inflammatory response [20,36,37]. Cytokines are crucial players in inflammatory-associated disorders such as NAFLD and are considered potential therapeutic targets. In the present study, we observed that APPLIVER and ACTEOS showed a significant reduction of TNF-α at both the gene and protein levels. TNF-α plays a pivotal role in the multi-step process of NASH’s development by inducing key enzymes of lipid metabolism, inflammatory cytokines, and fibrosis-associated proteins [37]. Hence, the attenuation of TNF-α’s action in the liver may help prevent or delay the development of NASH associated with metabolic syndrome. Although not directly on NASH model, the role of TA in attenuating the levels of proinflammatory cytokines has been demonstrated in chemical-induced inflammation in mice. In lipopolysaccharide (LPS)-induced neuroinflammation in mice, TA downregulated the expression of TNF-α and IL-1β by inhibiting nuclear factor-kappa B (NF-κB) and activating liver X receptor alpha (LXRα) receptors [29]. This was confirmed by the observation that TA decreased the serum IL-1β, IL-6, and TNF-α levels in acetaminophen-induced liver damage in mice by inhibiting the NF-κB and mitogen-activated protein (MAP) kinase activities [27].

Several reports have pointed out that an excess of FFA can also directly trigger mitochondrial dysfunction that leads to reactive oxygen species (ROS) production, another key hallmark of NASH [38,39,40]. ROS production has also been linked to hepatic inflammation by increasing the secretion of TNF-α from hepatocytes and Kupffer cells, thus upregulating the synthesis of inflammatory cytokines [35]. We therefore assessed the role of APPLIVER and ACTEOS in attenuating the production of ROS in FFA-overloaded hepatocytes. In line with our previous results, an FFA excess enhanced the production of ROS, similar to a level induced by hydrogen peroxide (the positive control). Cotreatment with APPLIVER resulted in an apparent but not significant antioxidant effect, while ACTEOS showed a significant reduction in ROS generation. A similar observation was reported in oxidative-stress-induced HepG2 and SH-SY5Y cell lines, implying a role for acteoside as a hepatoprotective and neuroprotective agent, respectively [33].

As mentioned earlier, the severity of hepatic fibrosis is the primary predictor of liver-related morbidity and mortality in NAFLD patients [7]. The role of HSCs in liver fibrogenesis is well established and occurs by the activation and alteration of genes involved in the turnover of extracellular matrix components [8]. We showed that excess FFA in hepatocytes activates the HSCs, indicating that cell-to-cell proximity between the two cell types is necessary for the initiation of the fibrotic process and overproduction of collagen type I [9]. ACTEOS was able to reduce both the gene expression and protein levels of COL1A1 in our SCC model of NASH. Acteoside was shown to be a potential renal and prostate fibrosis antagonist by inhibiting the TGF-β1/Smads signaling pathway in some studies [41,42].

APPLIVER at 96 h showed a non-significant reduction in COL1A1 gene expression, while the level of the collagen protein was significantly reduced. This discrepancy may be due to the differences in the transcription and translation rates of the COL1A1 gene and protein, as well as differences in their half-lives in achieving steady-state levels [43]. A recent study showed that TA inhibited platelet-derived growth-factor-BB-stimulated HSC activation, as evidenced by the inhibition of cell proliferation, migration, and colony formation, as well as a decreased expression of TGF-β and α-SMA through blocking the PI3K/Akt/mTOR and NF-κB signaling pathways [44]. The same study also revealed that TA decreased the expression of collagen types I and III, alleviating the excessive deposition of ECM.

Overall, the present study shows that the in vitro model provides a promising tool for investigating the efficacy of new candidate drugs, particularly on pathophysiological mechanisms involved in NASH (steatosis, inflammation, oxidative stress, fibrogenesis). The use of human cell lines in both monoculture and simultaneous co-culture setups facilitates standardized protocol and reproducible studies. However, caution is needed to export in vitro data to far more complex animal and human subjects. In addition, the role of other critical pathological events, such as apoptosis, insulin resistance, as well as the involvement of microbiota, should be considered in future studies. Nevertheless, the data obtained herein support the beneficial effects of APPLIVER and ACTEOS in the in vitro NASH environment.

In summary, our results reveal that APPLIVER and ACTEOS are promising therapeutic compounds in counteracting the pathophysiological mechanisms caused by an excess of FFA in the hepatocytes. ACTEOS provided solid results by reducing both the inflammation and oxidative stress, and, most importantly, attenuating collagen production elicited by the excess of FFA in the SCC model. In the absence of FFA, the use of APPLIVER or ACTEOS alone did not induce any cellular alteration in the processes under study, suggesting the safety of this compound in the absence of steatosis. The beneficial effects observed can be attributed to triterpenic acid and acteoside, since they are the main components of APPLIVER and ACTEOS, respectively; but the contribution of other components present in the extracts cannot be excluded. Taken together, the use of the data obtained herein using the in vitro NASH model remains to be explored before it can be used in the clinical NAFLD/NASH setting.

## 4. Materials and Methods

### 4.1. Compounds and Chemicals

APPLIVER is a new commercialized biotechnological compound (MRM01ECL, *Cydonia oblonga* extract, ABRESEARCH SRL, Brendola, Italy). It is an extract from the plant cell suspension culture of *Cydonia oblonga*’s fruit (quince). This extract was obtained by an alcoholic extraction of freeze-drying cells and containing 10.8% of Euscaphic acid, expressed as triterpenic acid (TA: MW: 488.7 g/mol). The solubility of the extract is 25 mg on 0.5 mL of DMSO. From 2 g of APPLIVER, 5.4 mg were weighted to prepare a 50 mg/mL stock solution in DMSO.

ACTEOS is a commercialized biotechnological compound (LCK03ECL, *Lippia citriodora* extract, ABRESEARCH SRL, Brendola, Italy). It is an extract from plant cell suspension culture of *Lippia citriodora* (or *Aloysia citriodora*) belonging to Verbenaceae family, containing 71.2% of phenylpropanoid expressed as acteoside (MW: 4624.59 g/mol). The solubility of the extract is 25 mg on 0.3 mL of 60%/40% water/ethanol. From 2 g of ACTEOS, 59.3 mg were weighed to prepare an 83.3 mg/mL stock solution in 60%/40% water/ethanol. For each experiment, dilutions of the stock solutions in cell culture media were freshly prepared every time.

Dulbecco’s modified Eagle’s high-glucose medium (DMEM-HG), L-glutamine, penicillin/streptomycin, and fetal bovine serum were purchased from Euroclone (Milan, Italy). 3-(4,5-dimethylthiazol-2-yl)-2,5-diphenyltetrazolium bromide (MTT), bicinchoninic acid solution-kit (B9643); bovine albumin Cohn V fraction (A4503); dimethyl sulfoxide (DMSO), hydrogen peroxide (H1009), Nile red (N3013), oleic acid (C18:1), palmitic acid (C16:0), phosphate-buffered saline (PBS) and tri-reagent (T9424) were obtained from Sigma Chemical (St. Louis, MO, USA). AlphaLISA cell lysis buffer and Alpha Immunoassay buffer were obtained from PerkinElmer (Boston, MA, USA). An iScript™ cDNA Synthesis kit (170-8890) and iQ SYBR Green Supermix (170-8860) were purchased from Bio-Rad Laboratories (Hercules, CA, USA). 2′,7′-dichlorodihydrofluorescein diacetate (H_2_DCFDA) (D399) was obtained from Molecular Probes (Milan, Italy).

### 4.2. Cell Culture

Hepatic stellate cells (HSCs) LX2 were kindly provided by S. L. Friedman (Mount Sinai School of Medicine, New York, NY, USA) while hepatoma-derived cell line Huh7 (JHSRRB, Cat#JCRB0403) was obtained from the Japanese Health Science Research Resource Bank (Osaka, Japan). LX2 cells express α-SMA under all culture conditions and therefore must be regarded as at least partially activated. Despite this, LX-2 cells can be quiesced by growth in matrigel or low serum [45,46]. Therefore, LX2 cells were maintained in DMEM high glucose (HG) supplemented with 100 U/mL penicillin/streptomycin, 2 mM L-glutamine, and 1% *v*/*v* fetal bovine serum at 37 °C in 5% CO_2_ air humidified atmosphere.

The simultaneous co-culture was prepared in a cell ratio of 5:1 (hepatocytes:HSCs), resembling the in vivo ratio of parenchymal to non-parenchymal cells in the liver [47]. For each experiment, SCC was maintained in DMEM-HG supplemented with 100 U/mL penicillin/streptomycin, 2 mM L-glutamine, and 1% *v*/*v* fetal bovine serum. LX2 cells were maintained throughout in 1% FBS while Huh7 cells were first adapted to 1% FBS medium by gradually reducing the medium’s serum concentration two weeks before the experiment. For the induction of NASH, SCC and monoculture were exposed to 1200 µM of free fatty acids (FFA) (oleic:palmitic ratio 2:1 μmol/µmol) as previously described by our group [9,18]. Media containing FFA and APPLIVER and ACTEOS were refreshed every 2 days until they reached the experimental time point. Effects of APPLIVER and ACTEOS were evaluated also in absence of FFA to assess possible side effects. For each culture setup (monoculture or SCC), cells exposed to the vehicle were used as a control. The maximal concentration of vehicle (DMSO) was 0.22% *v*/*v*. SCC total cell densities for each time point were 20,000 cells/cm^2^, 10,000 cells/cm^2^, and 5000 cells/cm^2^ at 24 h, 96 h, and 144 h, respectively. A summary of the experimental setup, conditions, and experimental checkpoints are described in Figure 8 below.

### 4.3. Experimental Dose Determination and Cell Viability Assay

The cytotoxic effect of FFA (1200 µM), APPLIVER, and ACTEOS alone or in combination was assessed by MTT colorimetric assay at 24, 96, and 144 h. Monoculture and SCC were plated at a cell density as described before [36] and treated accordingly with increasing concentrations of APPLIVER and ACTEOS. Likewise, cells were also exposed to control medium with increasing concentrations of APPLIVER and ACTEOS only to assess their intrinsic toxicity in the system without FFA. After 24 h, the medium was removed, and cells were incubated for 1 h with MTT at a concentration of 0.5 mg/mL. Afterward, the culture medium was removed, and formazan crystals were dissolved in 200 µL of DMSO. A volume of 100 µL from each well was transferred to a microtiter plate, and the optical density (OD) was determined at 562 nm wavelength on an Enspire^®^ Multimode Plate Reader (PerkinElmer, Waltham, MA USA).

To identify the experimental concentration for each compound, we followed the recommendation of ISO 10993-5 (in vitro cytotoxicity test on extracts) [20]. A reduction in cell viability of more than 20% was considered as the exclusion criteria. The cutoff concentration meeting this requirement was selected as the experimental concentration for each compound.

### 4.4. Fluorimetric Determination of Intracellular Fat Content—Nile Red Staining

Intracellular fat content was determined by flow cytometry using Nile red staining, a vital lipophilic dye used to label fat accumulation in the cytosol [21]. After 24 and 96 h of FFA exposure, adherent monolayer cells were washed twice with PBS and detached by trypsinization. After 5 min of centrifugation at 1500 rpm, the cell pellet was resuspended in 3 mL of PBS and incubated with 0.75 μg/mL Nile red dye for 15 min at room temperature. Nile red intracellular fluorescence was detected using a Becton Dickinson FACSCalibur system on the FL2 emission channel through a 585 ± 21 nm bandpass filter, following excitation with an argon-ion laser source at 488 nm. Data were collected in 10,000 cells and analyzed using Cellquest software from BD Biosciences (San Jose, CA, USA).

### 4.5. RNA Extraction, cDNA Synthesis, and Gene Expression Analysis by qRT-PCR

Total RNA was extracted from culture harvest using a tri-reagent kit (SIGMA) according to the manufacturer’s instructions. The total RNA concentration was then quantified spectrophotometrically at 260 nm in a Beckman CoulterDU^®^ 730 spectrophotometer (Fullerton, CA, USA), while purity was evaluated by measuring the A260/A280 ratio. Total RNA (1 μg) was reverse transcribed using the High-Capacity cDNA Reverse Transcription Kit (Applied Biosystems, Waltham, MA, USA) according to the manufacturer’s protocol in a thermal cycler (GeneAmp PCR System 2400, PerkinElmer, Boston, MA, USA). Quantitative PCR was performed in i-Cycler IQ (Bio-Rad, Hercules, CA USA). All primer pairs were designed using the software Beacon Designer 8.12 (PREMIER Biosoft International Palo Alto, CA, USA) and were synthesized by Sigma Genosys Ltd. (London Road, UK). Primer sequences are specified in Table 2 HPRT and 18S were used as reference genes. PCR amplification was performed in 20 µL reaction volume containing 25 ng of cDNA, 1X iQ SYBR Green Supermix, and 250 nM gene-specific sense and antisense primers and 100 nM primers for 18S. The data were analyzed using Bio-Rad iQ5 software (version 3.1).

### 4.6. Intracellular ROS Generation by H_2_DCFDA

Intracellular reactive oxygen species (ROS) generation was measured using the cell-permeable fluorogenic substrate H_2_DCFDA. This non-fluorescent probe is easily taken up by cells, and, after intracellular cleavage of the acetyl groups, is trapped and may be oxidized to the fluorescent compound 2′,7′-dichlorofluorescein (DCF; the monitored fluorophore) by intracellular ROS. Huh7 cells were exposed for 1 h with APPLIVER and ACTEOS, FFA co-treated with APPLIVER and ACTEOS, hydrogen peroxide as the positive control, and vehicle control. Then cells were washed with PBS and incubated for 30 min with FBS-free culture medium supplemented with 25 mM HEPES/20 µM H_2_DCFDA at 37 °C. Afterward, cellular fluorescence was quantified by scanning the signal of the attached cells in each well with an Enspire^®^ Multimode Plate Reader (PerkinElmer) at excitation wavelength 505 nm and emission 525 nm. Fluorescence was normalized by the total proteins present in the cell lysates (μg) assessed using fluorescamine for each sample.

### 4.7. Quantification of COL1A1 Deposition (AlphaLISA Quantification)

The amount of collagen present in the SCC system was quantified by AlphaLISA (PerkinElmer AL371), according to the manufacturer’s protocol. Briefly, after 96 and 144 h of treatment, cells were washed with PBS and lysed with AlphaLISA lysis buffer. Cell lysates were diluted 1:10 in AlphaLISA Immunoassay Buffer, and 2 µL of samples were incubated for 30 min at 23 °C with 4 µL of 5X AlphaLISA Anti-hCOL1A1 acceptor beads. Subsequently, 4 µL of 5X Biotinylated Anti-hCOL1A1 antibody was added and incubated for 60 min. Then 10 µL of 2X SA-Donor Beads were added to each sample and incubated for another 30 min in the dark. The signal reading was performed using an EnVision-Alpha Reader at 615 nm. Collagen concentration was calculated by extrapolation of the data in a standard curve constructed with known concentrations of collagen.

### 4.8. Normalization vs. µg of Total Proteins

To obtain more accurate data and to exclude misleading results due to eventual differences in the number of attached cells, data from collagen quantification, ROS assay, and AlphaLISA assay were normalized by µg of total proteins in each well. AlphaLISA lysis buffer interfered with the normal techniques for protein quantification (BCA, Bradford, 280 nm absorbance); thus, the total protein concentration was calculated fluorometrically (ex: 390 nm–em: 460 nm) by fluorescamine reaction. Total proteins were quantified by fluorescamine, a nonfluorescent molecule that reacts readily with primary amines in amino acids and peptides to form stable, highly fluorescent compounds. Ten microliters of fluorescamine at 4 mg/mL in DMSO were mixed with 40 μL of cell lysate diluted 1:2 *v*/*v*. Fluorescence was read at 460 nm (excitation wavelength of 390 nm) in an EnSpire^®^ Multimode Plate Reader (PerkinElmer, Waltham, MA USA). Total protein was calculated using a BSA standard curve dilution.

### 4.9. TNF-α ELISA and Bicinchoninic Acid (BCA) Assay

The TNF-α release in cell culture supernatant was quantified by human TNF alpha SimpleStep ELISA^®^ Kit (ABCAM, AB181421), according to the manufacturer’s protocol. Briefly, after 1 h of treatment, cell culture supernatants were collected. Volumes of 50 μL of samples or standard were loaded to appropriate wells. After which, 50 μL of the antibody cocktail were added to each well. The plate was incubated for 1 h at room temperature on a plate shaker set to 400 rpm. The plate was then washed 3 times with 50 μL 1X Wash Buffer PT in each well. Then, 100 μL of TMB development solution were added to each well and incubated for 10 min in the dark on a plate shaker set to 400 rpm. Finally, 100 μL of stop solution in each well were added; the absorbance was read at 450 nm. TNF-α (pg/μL) concentration was calculated by extrapolation of the data in a standard curve constructed with known concentrations of TNF-α.

To obtain more accurate data and to exclude misleading results due to eventual differences in the amount of proteins released in the supernatant, data from TNF-α ELISA were normalized by µg of total proteins in each well. Total protein was calculated by bicinchoninic acid (BCA) assay using a BSA standard curve dilution.

### 4.10. Statistical Analysis

All values are presented as means  ±  standard deviation (S.D.) from three independent experiments. Statistical analysis was performed by one-way ANOVA (with post hoc test) or Student’s *t*-test, whenever applicable using GraphPad Prism, version 5.0 (GraphPad Software, San Diego, CA, USA) (v5). Statistical significance was determined at *p*  <  0.05.

## 5. Conclusions

Overall, the data obtained in our well-established in vitro model of NASH point to beneficial effects of APPLIVER and ACTEOS in counteracting the deleterious effects of FFA. Among the two, ACTEOS provided promising results by reducing both the inflammation and the oxidative stress and, most importantly, attenuating the collagen production elicited by the excess of FFA in the SCC model. The beneficial effects observed can be attributed to triterpenic acid and acteoside, since they are the main components of APPLIVER and ACTEOS, respectively; but the contribution of other components present in the extracts cannot be excluded. The use of the data obtained in vitro to the clinical NAFLD/NASH setting remains to be explored.

## Figures and Tables

**Figure 1 ijms-23-03562-f001:**
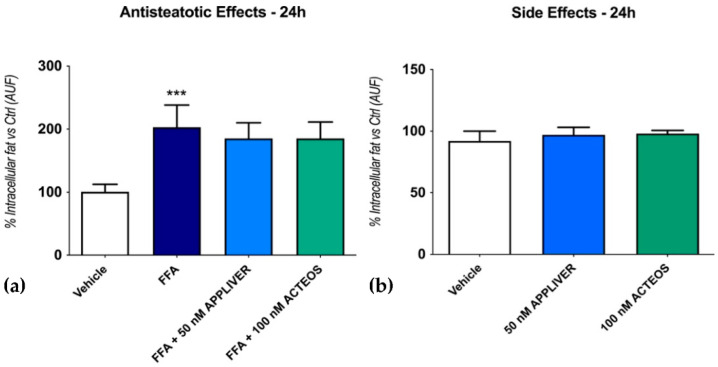
Assessment of the effect of APPLIVER and ACTEOS as antisteatotic (**a**) and its side effects (**b**) after 24 h on hepatocytes. Values presented are the mean ± SD of three independent experiments. *** *p* < 0.001 vs. vehicle. Abbreviations: AUF, arbitrary unit of fluorescence; FFA, free fatty acids.

**Figure 2 ijms-23-03562-f002:**
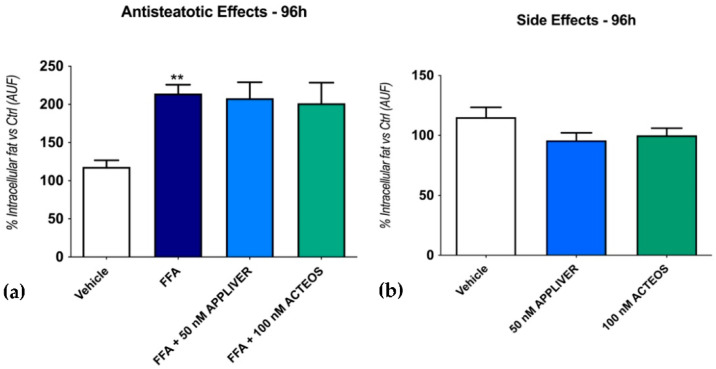
Assessment of the effect of APPLIVER and ACTEOS as antisteatotic (**a**) and its side effects (**b**) after 96 h on SCC. Values presented are the mean ± SD of three independent experiments. ** *p* < 0.01 vs. vehicle. Abbreviations: AUF, arbitrary unit of fluorescence; FFA, free fatty acids.

**Figure 3 ijms-23-03562-f003:**
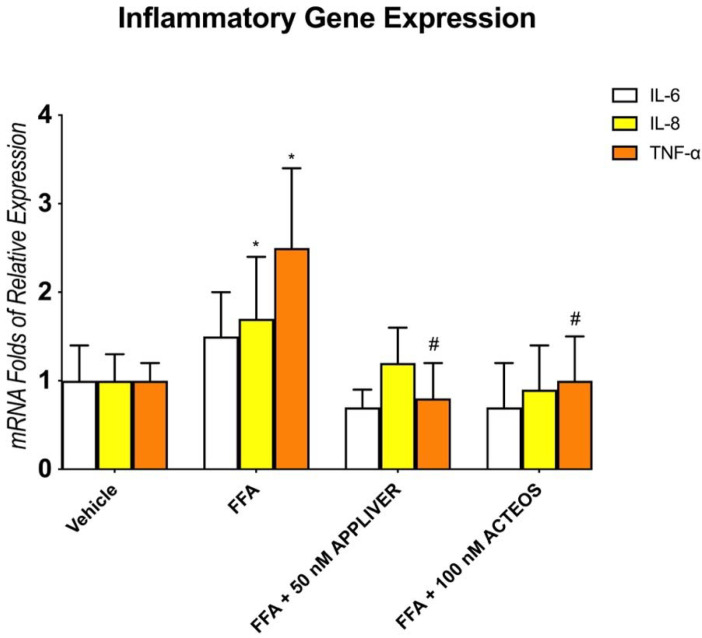
Assessment of anti-inflammatory properties of APPLIVER and ACTEOS on inflammatory cytokines mRNA expression. Gene expression analysis of pro-inflammatory cytokines (folds of mRNA expression vs. vehicle). Values presented are the mean ± SD of three independent experiments. * *p* < 0.05 vs. vehicle, # *p* < 0.05 vs. FFA. Abbreviations: FFA, free fatty acids; IL-6, interleukin-6; IL-8, interleukin-8; TNF-α, tumor necrosis—alpha.

**Figure 4 ijms-23-03562-f004:**
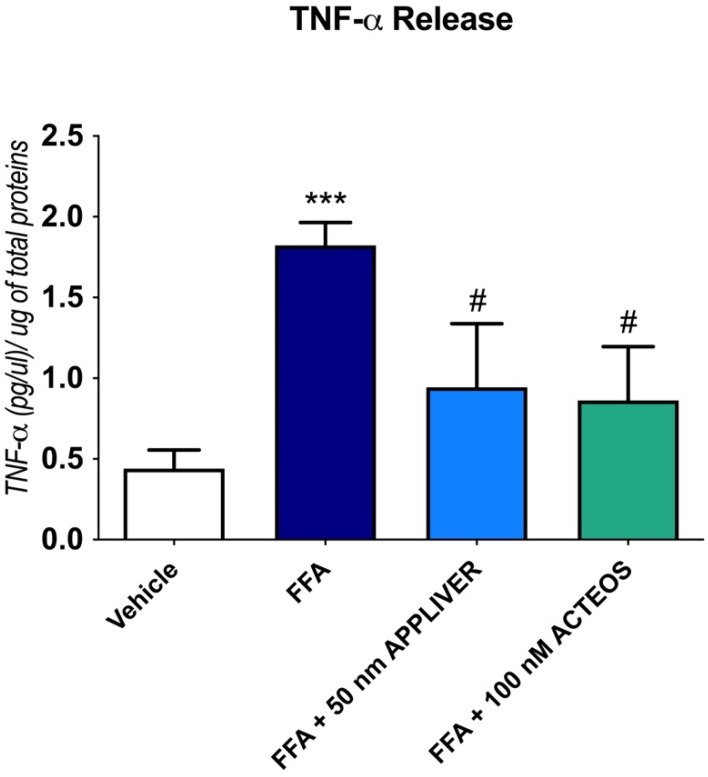
Quantification of TNF-α release in cell culture supernatant ((pg/µL)/µg of total proteins) upon treatment with either APPLIVER (A) or ACTEOS (B) ± FFA after 1 h. Values presented are the mean ± SD of three independent experiments. *** *p* < 0.001 vs. vehicle, # *p* < 0.05 vs. FFA. Abbreviations: FFA, free fatty acids; TNF-α, tumor necrosis—alpha.

**Figure 5 ijms-23-03562-f005:**
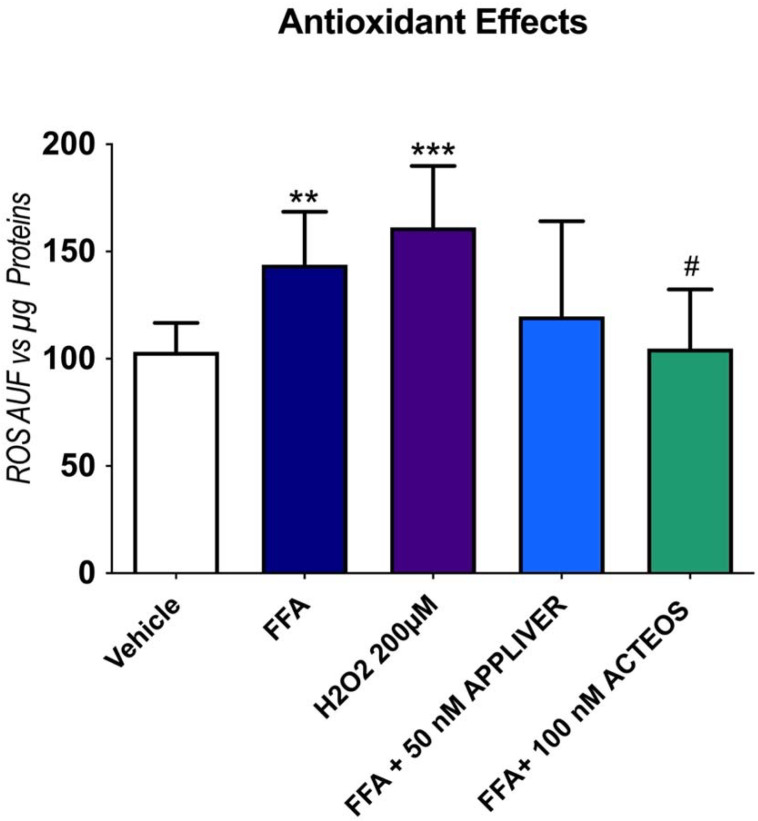
Assessment of antioxidant properties of APPLIVER and ACTEOS induced by FFA. Fluorescence was normalized by the total proteins present in the cell lysates (μg) assessed using fluorescamine assay. Values presented are the mean ± SD of three independent experiments. ** *p* < 0.01 vs. vehicle, *** *p* < 0.001 vs. vehicle, # *p* < 0.05 vs. FFA. Abbreviations: AUF, arbitrary unit of fluorescence; FFA, free fatty acids; H_2_O_2_, hydrogen peroxide.

**Figure 6 ijms-23-03562-f006:**
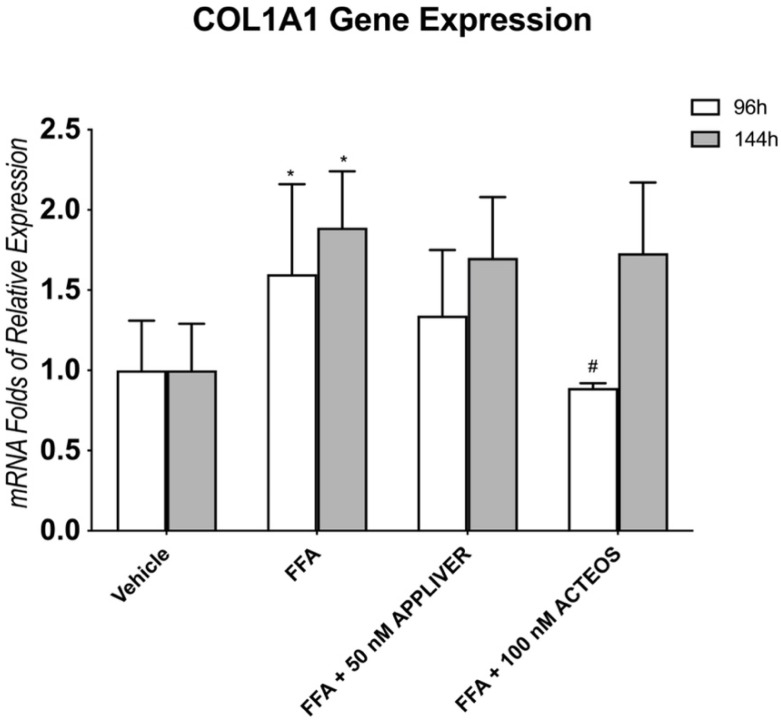
Gene expression of COL1A1 upon exposure to FFA and APPLIVER and ACTEOS after 96 h and 144 h. Values presented are the mean ± SD of three independent experiments. * *p* < 0.05 vs. CTRL, # *p* < 0.05 vs. FFA. Abbreviations: COL1A1, collagen type I alpha 1 chain; FFA, free fatty acids.

**Figure 7 ijms-23-03562-f007:**
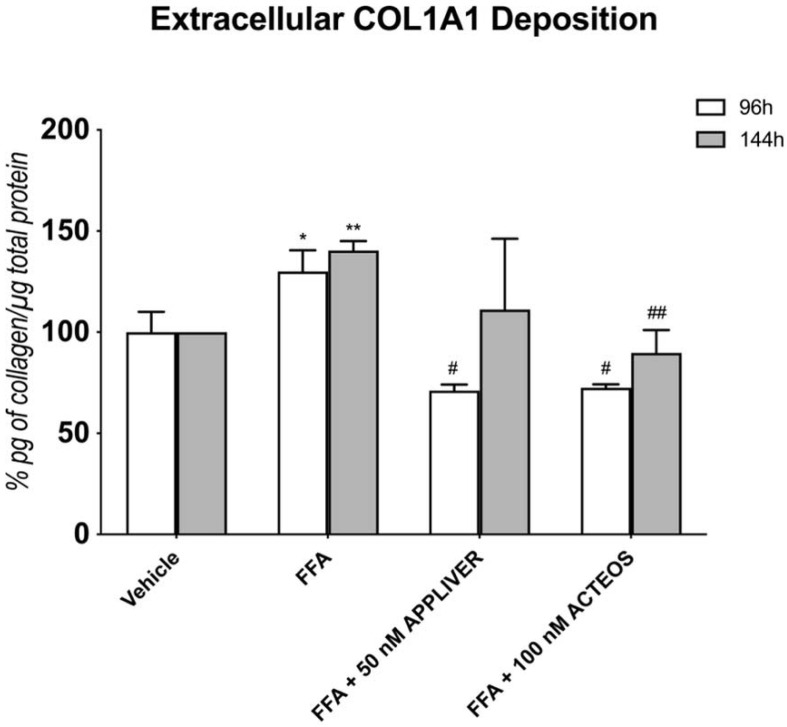
Quantification of extracellular COL1A1 deposition (% pg of collagen/µg total protein) using ELISA upon treatment with either APPLIVER or ACTEOS ± FFA after 96 h and 144 h. COL1A1 deposition was normalized to the total proteins in the cell lysate and reported as percentage vs. vehicle. Values presented are the mean ± SD of three independent experiments. * *p* < 0.05 vs. vehicle, ** *p* < 0.001 vs. vehicle, # *p* < 0.05 vs. FFA, ## *p* < 0.01 vs. FFA. Abbreviations: COL1A1, collagen type I alpha 1 chain; FFA, free fatty acids.

**Figure 8 ijms-23-03562-f008:**
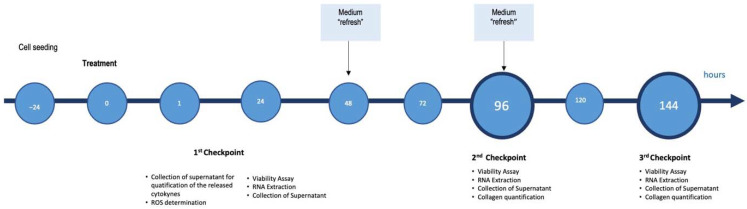
Scheme of the culture proceedings and the experimental checkpoints with the relative determinations.

**Table 1 ijms-23-03562-t001:** Cell viability after APPLIVER and ACTEOS exposure with or without FFA (% vs. vehicle).

Treatment		Huh7 Monoculture	Huh7—LX2 Simultaneous Co-Culture	
		24 h	96 h	144 h
		Mean ± SD	Mean ± SD	Mean ± SD
**+FFA**	Vehicle	100 ± 0	100 ± 0	100 ± 0
	FFA	99 ± 11	86 ± 7	91 ± 5
	APPLIVER 10 nM	89 ± 18	83 ± 2	97 ± 5
	APPLIVER 50 nM	94 ± 22	69 ± 28 *	111 ± 8
	ACTEOS 100 nM	98 ± 6	81 ± 9	90 ± 3
	ACTEOS 10,000 nM	96 ± 11	75 ± 7 *	88 ± 1
**−FFA**	Vehicle	100 ± 0	100 ± 0	100 ± 0
	APPLIVER 10 nM	99 ± 17	99 ± 3	118 ± 37
	APPLIVER 50 nM	90 ± 16	82 ± 1	105 ± 21
	ACTEOS 100 nM	93 ± 12	82 ± 3	96 ± 20
	ACTEOS 10,000 nM	82 ± 7	69 ± 7 *	71 ± 6 *

The cell viability was determined by MTT assay. Values presented are the mean ± SD of three independent experiments. * 80–60% cell viability = weakly cytotoxic. Abbreviations: FFA, free fatty acids; Huh7, human hepatoma cell line; LX2, human hepatic stellate cell line; nm, nanomolar.

**Table 2 ijms-23-03562-t002:** Primer Sequences.

Gene Name	Accession Number	Forward	Reverse
IL-8	NM_000584	GACATACTCCAAACCTTTCCAC	CTTCTCCACAACCCTCTGC
IL-6	NM_000600	ACAGATTTGAGAGTAGTGAGGAAC	GGCTGGCATTTGTGGTTGG
TNF-α	NM_000594	GTGAGGAGGACGAACATC	GAGCCAGAAGAGGTTGAG
COL1A1	NM_000088	CGGAGGAGAGTCAGGAAG	ACACAAGGAACAGAACAGTC
18S	NR_003286.2	TAACCCGTTGAACCCCATT	CCATCCAATCGGTAGTAGCG
HPRT	NM_000194.	ACATCTGGAGTCCTATTGACATCG	CCGCCCAAAGGGAACTGATAG

Abbreviations: COL1A1, collagen type I A1; HPRT, hypoxanthine phosphoribosyltransferase; IL-6, interleukin-6; IL-8, interleukin-8; TNF-α, tumor necrosis—alpha.

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
