# Peer review of "The Beneficial Effects of Triterpenic Acid and Acteoside in an In Vitro Model of Nonalcoholic Steatohepatitis (NASH)"

_ijms, 2022, doi:10.3390/ijms23073562_

Round 1

Reviewer 1 Report

Authors reported that the efficacy of triterpenic acid and acteoside on 19 pathophysiological mechanisms related to NASH.

  1. Instead of Ref. 4, Add reference as follows: Eslam M, et al. The Asian Pacific Association for the Study of the Liver clinical practice guidelines for the diagnosis and management of metabolic associated fatty liver disease. Hepatol Int. 2020 Dec;14(6):889-919.
  2. In lines 9, 55, 72, “Triterpenic acid (TA) and…”?
  3. In line 161, what is “[22,23],[24]. In line with”?
  4. In Table 1, all Figures and Figure legends, please add cell names.
  5. Authors should examine or discuss about the role of apoptosis in NASH development (See: Kanda T, et al. World J Gastroenterol. 2018 Jul 7;24(25):2661-2672.).
  6. Authors should ask their manuscript before resubmission.

Author Response

REVIEWER 1

Authors reported that the efficacy of triterpenic acid and acteoside on pathophysiological mechanisms related to NASH.

1. Instead of Ref. 4, Add reference as follows: Eslam M, et al. The Asian Pacific Association for the Study of the Liver clinical practice guidelines for the diagnosis and management of metabolic associated fatty liver disease. Hepatol Int. 2020 Dec;14(6):889-919

Response: the suggested reference was used in the revised manuscript.

2. In lines 9, 55, 72, “Triterpenic acid (TA) and…”?

Response: point has been addressed in the manuscript. Capital “A” in “acid” was changed to lower case.

3. In line 161, what is “[22,23],[24]. In line with”?

Response: Line 161 citing references [22,23,24] refers to the summarized peak concentrations of TNF-alpha in the plasma/cell supernatant after 1-2 hours upon induction of inflammation. The only difference between these three research papers are the types of inflammation models used. Reference 22 used endotoxin to induce TNF-alpha release in mononuclear cells, reference 23 used lipopolysaccharide (LPS), and reference 24 used RMP16, a recombinant TNF α-derived polypeptide.

4. In Table 1, all Figures and Figure legends, please add cell names.

Response: point has been addressed in the Tables. Abbreviations were also defined.

5. Authors should examine or discuss about the role of apoptosis in NASH development (See: Kanda T, et al. World J Gastroenterol. 2018 Jul 7;24(25):2661-2672.).

Response: Thank you for the suggestion. We are aware that lipotoxicity can trigger apoptosis and play a role in the development of NASH. Our model will not allow studying this aspect as we examined steatosis, inflammation, oxidative stress, and fibrogenesis. Nevertheless, in the revised discussion we added a line that stresses the role of apoptosis in NASH development and thus, deserves investigation in future studies.

6. Authors should ask their manuscript before resubmission.

Response: Thank you for pointing out this. If you are referring to the author agreement statement, the manuscript has been read, edited, and approved by all named authors.

Reviewer 2 Report

I read with interest the manuscript by Salvoza and colleagues on the beneficial effects of triterpenic acid and acteoside in vitro. The authors concluded that these two compounds deserve further investigation for possible use in NASH treatment.

The manuscript is well written and all the experiments are clearly exposed.

My major concern regards the in vitro approach to mimic NASH condition. The liver is characterized by a complex architecture and the interaction of different type of cells is a common phenomenon; reproducing these settings in in vitro experiments can be a reliable way to investigate the cell-cell interaction in the progression of disease. Why the authors choosed to use HSC and not kuppfer cells to assess the inflammatory response? 

Author Response

REVIEWER 2

I read with interest the manuscript by Salvoza and colleagues on the beneficial effects of triterpenic acid and acteoside in vitro. The authors concluded that these two compounds deserve further investigation for possible use in NASH treatment.

The manuscript is well written and all the experiments are clearly exposed.

My major concern regards the in vitro approach to mimic NASH condition. The liver is characterized by a complex architecture and the interaction of different type of cells is a common phenomenon; reproducing these settings in in vitro experiments can be a reliable way to investigate the cell-cell interaction in the progression of disease. Why the authors choosed to use HSC and not kuppfer cells to assess the inflammatory response? 

Response:

We thank you for the comments. We are aware that the development of NASH involves the interplay of different factors and cell types. In this in vitro study, we utilized human cell lines in both monoculture and co-culture setups. These set-ups facilitate standardized protocol and reproducible studies. As mentioned in the discussion section, Kupffer cells are important drivers of hepatic inflammation. However, hepatocytes per se have a role in initiating the inflammatory response known as lipototoxicity, as  confirmed in  our previous study cited in the manuscript. Therefore, the gene expression and the supernatant release of TNF-alpha were determined using hepatocytes overloaded with FFA. The details of the set-up are fully described in the method section.

In the present study, we explored the importance of cell-to-cell proximity between hepatocytes and HSCs to study fibrogenesis, not inflammatory response. To support this, the following lines were pointed out in the discussion section of the manuscript: “The role of HSCs in liver fibrogenesis is well established and occurs by the activation and alteration of genes involved in the turnover of extracellular matrix components [8]. We showed that excess FFA in hepatocytes activate the HSCs, indicating that cell-to-cell proximity between the two cell types is necessary for the initiation of the fibrotic process and overproduction of collagen type I.”

Round 2

Reviewer 1 Report

All queries have been addressed.